# Developing and Validating an Individual-Level Deprivation Index for Children’s Health in France

**DOI:** 10.3390/ijerph192416949

**Published:** 2022-12-16

**Authors:** Remi Laporte, Philippe Babe, Elisabeth Jouve, Alexandre Daguzan, Franck Mazoue, Philippe Minodier, Guilhem Noel, Diego Urbina, Stephanie Gentile

**Affiliations:** 1Permanence d’Accès aux Soins de Santé Mère-Enfant, Hôpital Nord, Assistance Publique-Hôpitaux de Marseille, 13005 Marseille, France; 2Equipe de Recherche EA 3279 “Santé Publique, Maladies Chroniques et Qualité de Vie”, Faculté de Médecine, Aix Marseille University, 13005 Marseille, France; 3Service d’Accueil des Urgences Pédiatriques, Hôpital Nord, APHM, 13005 Marseille, France; 4Permanence d’Accès aux Soins de Santé Pédiatrique, Hôpitaux Pédiatriques de Nice CHU-Lenval, 06200 Nice, France; 5Service d’Accueil des Urgences Pédiatriques, Hôpitaux Pédiatriques de Nice CHU-Lenval, 06200 Nice, France; 6Service d’Evaluation Medicale, Assistance Publique-Hôpitaux de Marseille, 13005 Marseille, France

**Keywords:** social determinants of health, screening, deprivation, health inequalities, social support, access to health care

## Abstract

Background: Deprivation generates many health inequalities. This has to be taken in account to enhance appropriate access to care. This study aimed to develop and validate a pediatric individual-level index measuring deprivation, usable in clinical practice and in public health. Methods: The French Individual Child Deprivation Index (FrenChILD-Index) was designed in four phases: item generation then reduction using the literature review and expert opinions, and index derivation then validation using a cross-sectional study in two emergency departments. During these last two phases, concordance with a blinded evaluation by an expert enabled us to determine thresholds for two levels of moderate and severe deprivation. Results: The generation and reduction phases retained 13 items. These were administered to 986 children for the derivation and validation phases. In the validation phase, the final 12 items of the FrenChILD-Index showed for moderate deprivation (requiring single specific care for deprived children) a sensitivity of 96.0% [92.6; 98.7] and specificity of 68.3% [65.2; 71.4]. For severe deprivation (requiring a multidisciplinary level of care), the sensitivity was 96.3% [92.7; 100] and specificity was 91.1% [89.2; 92.9]. Conclusions: The FrenChILD-Index is the first pediatric individual-level index of deprivation validated in Europe. It enables clinical practice to address the social determinants of health and meet public health goals.

## 1. Introduction

Social determinants of health (SDH) have a tremendous impact on children, generating early health inequalities. Pillas et al. widely reviewed the impact of SDH on every organ during childhood including over 201 studies in European countries [1]. Pearce et al. highlighted the long-lasting impact of SDH through the increase in overweight, poor mental well-being, longstanding illnesses, asthma, and unintentional injuries [2]. The impact of SDH has recently increased on the mental health of children and adolescents including sleep disorders [3,4] and an impact on schooling [5]. Early health inequalities are also reinforced by barriers in access to health care determining vicious circles in therapeutic and preventive care [6,7].

Deprivation is multidimensional and can affect every SDH domain: economic stability, but also education, health and health care, the neighborhood and built environment (including housing), social and community context, and family context [8,9,10,11].

Health systems have implemented many actions to tackle SDH, although a global action of health in all policies remains known as necessary [12]. In mainstream health care systems in France, health care is paid for by various social security systems. However, patients must already have followed an administrative registration so that their medical consultations, care, and medication can be paid for. Since 1998, “Permanences d’Accès aux Soins de Santé” (PASS) are specific socio-medical units implemented in hospitals to enhance access to health care for severely deprived patients with complex needs [13,14]. These units must interface with the mainstream health care system by screening patients in need of their care to ensure the respect of proportionate universalism in access to health care [15,16,17].

It is common for most pediatricians in mainstream health care system, regardless of their clinical practice settings, to provide care to deprived children, given the significant prevalence of economically poor children (21.0% in France, 2018) [15,18]. Screening for deprivation is crucial to provide appropriate referral and support for children [19], and pediatricians often need assistance with this [20].

Area-based deprivation indices have been implemented to compare deprivation within different geographic areas, but were limited to the study of individual health care pathways. Several studies have described health inequalities over frequent diseases in Europe for children in epidemiological studies [6,21,22]. However, strong variations in precision were found between area-based indices [23] and between the components of an index [24]. Significant discrepancies were also found when compared with an individual-level index [25]. Zelenina et al. are currently involved in reviewing their properties [26].

However, most terms in these indices (the rate of unemployment, of tenants, of single parenthood in an area…) are unavailable during care, and poorly reliable to individual situations of care.

Several individual-level indices have been implemented in France. The social handicap index uses 111 items and could not be realized in daily clinical practice [27]. Pascal et al. only included questions about health insurance coverage and income [28]. EPICES had some inappropriate questions during child health care related to lifestyle confounding factors about leisure activities (going to shows or on vacation) [29]. This variability was even more feared following COVID. Fouchard compared three adult individual-level deprivation indices and showed restrictions about their metrological performances [30]. Another index was designed for pregnancy, but most items were very specific to this status [31].

In children, Sokol reviewed 11 indices derived in the USA. Only three had their validity assessed [8]. All of these included items about adverse childhood experiences or parental depression. However, screening for deprivation (socially considered to be a condition for assistance) and addressing adverse childhood experiences (socially banned and legally prohibited) should be considered separately because the latter are deemed to be judgmental and cause response biases [32]. The WHO has also separated chapters of its strategy on social determinants of health reduction and the prevention of adverse childhood experiences [33].

Nicolas et al. used informal interviews to identify different situations of absolute precariousness requiring urgent social measures, recognized and well-managed precariousness, unrecognized precariousness and/or complicated by a significant deterioration in quality of life, and psychological vulnerability. However, the validity of these interviews was not evaluated [34].

Thus, to our knowledge, no convenient individual-level index to assess the global burden for child SDH deprivation has been validated in France or in Europe.

PASS professionals were highly skilled in determining the patients’ degree of deprivation and propose appropriate health care. However, screening mainstream patients would hamper their work. We used this situation and the skills of PASS professionals to define several degrees of deprivation, according to the types of social support and health care required [35]:

Patients with severe deprivation were those requiring multidisciplinary care in PASS. Multidisciplinary care combine at least two specific health care types offered in PASS among: social assessment (incl. to gain access to health insurance), free medical consultations, free medicine dispensing, physical accompaniment in health care, home visits for unhealthy housing, and multidisciplinary care coordination meetings [13,14].

Patients with moderate deprivation (e.g., independent enough to seek help and follow simple administrative procedures) were defined as those requiring only a single type of specific health care required for deprived children (the same list as above).

The aim of this study was to develop and validate a pediatric individual-level index for deprivation that is usable in clinical practice and public health: the French Child Individual-Level Deprivation Index (FrenChILD-Index).

## 2. Materials and Methods

The study design included four phases: item generation, item reduction, index derivation, and index validation.

For the item generation phase, a literature review was performed in October 2013, updated to September 2016, to extract the existing screening instruments for investigating deprivation, vulnerability, or poverty at the individual level, in PubMed (for main international references) and CAIRN (increasing exhaustiveness to articles in French). In addition, face-to-face semi-structured interviews of a heterogeneous sample of 13 senior experts in specific health care required for deprived children collected their opinions on the SDH domains to be measured, and on the complementary items to address. Interviews ended when data saturation was achieved. Data saturation was empirically estimated when the investigator found that the two last interviews did not add any new item or new themes. This was confirmed by two other investigators analyzing the interview content [36].

For the item reduction phase, a steering committee was set up. Eight of the 13 senior experts accepted our invitation to participate (one pediatrician, two nurses, two social workers, two health care and socio-educational managers, and one health mediator). Oral consensus was obtained using the nominal group method [37]. They selected relevant and acceptable items covering each SDH domain, and proposed their wording and informative appendix.

An independent reading group (one general practitioner, two pediatricians, three nurses and two social workers) reviewed the index and appendix elaborated by the steering committee. They validated the choice and wording for each item and appendix via an online questionnaire with a semi-quantitative rating of their relevance (discrete numerical scale: 1 to 9). Agreement of the reading group was calculated according to the analysis rules [38]. According to the ratings distribution, a table determined how each proposal was: appropriate, inappropriate, or uncertain; with a degree of agreement: strong/relative/indecision/lack of consensus.

For the index derivation and validation phases, 13 items judged as being appropriate or uncertain were tested on a sample of children.

A convenient sample was recruited in a cross-sectional multicentric study conducted between April 2018 and October 2019, in two French university hospitals with different rates of economic poverty: Marseille (1) and Nice (2). For this study of tool development and validation, sample representativeness was not controlled. Any child was eligible when aged 3 to 15 years old, when admitted to pediatric emergency units without life-threatening conditions or ongoing medico-legal procedures, outside the evenings, weekends, and peak periods. Children could only be selected once. Only one sibling was included. Informed consent was obtained from the participant’s legally authorized representative and the child themselves when aged 8 years old or above. For non-French speaking children, a professional telephone interpretation service was used (ISO 13611:2014; 17100:2015).

The primary outcome was the concordance between the FrenChILD-Index results and a blinded expert evaluation:

The FrenChILD-Index questions were asked by health care professionals untrained in its use.

Experts were trained health care professionals working in PASS (social worker, nurse or pediatrician; with at least one year of experience caring for deprived children). They assessed proxies for the global deprivation burden, blinded to the results of the FrenChILD-Index, with several criteria: (1) the type and number of SDH domains affected by deprivation [8,9]; (2) the type and amount of specific health care required for deprived children (both lists mentioned in the introduction); and (3) the need for admission to PASS (when at least two different types of specific health care required for deprived children were required) [13,35]. Deprivation was defined in each SDH domain as the shortage of assets or services needed to satisfy basic needs, in comparison with the current social standards, according to the French social evaluation framework [13,39,40].

A sample size of approximately 1000 children was required to show a sensitivity and specificity of 80% with a 95% confidence interval (CI) and a precision of 5% for primary outcome, considering an expected prevalence of deprivation for at least 25% (one hospital being in a deprived neighborhood, as found in a previous study by Bouhamam et al. [7]). A complete-case analysis was conducted to ensure that there was no information bias of any estimator nor reference evaluation, thus patients missing any value were excluded.

For the index derivation phase, item-internal consistency was assessed by correlating each FrenChILD-Index item with the SDH domain and health care need they were logically related to. Items were retained if they had a significant moderate correlation with the expected SDH domain affected by the deprivation questioned (Pearson correlation coefficient |r| ≥ 0.3).

Retained items were included in the multivariate analysis. We performed an exploratory factorial analysis using multiple correspondence analysis and retained eigenvalues >1. Internal consistency of the index was assessed using the Kuder-Richardson Formula 20 coefficient (K20, equivalent to the Cronbach alpha for dichotomous items [41,42,43]). For derivation of the index, a linear multiple regression weighting was chosen to maximize the predictivity with respect to the progressiveness in global deprivation burden [44]. This was assumed to be assessable by the proxies: the number of SDH domains affected by deprivation and the amount of specific health care required for deprived children. We performed two linear multiple regressions to test the retained items for predicting the selected proxies of global deprivation burden. Independent variables included all retained items for the FrenChILD-Index. Dependent variables were respectively: (i) the number of SDH domains affected by deprivation and (ii) the amount of specific health care required for deprived children. For FrenChILD-Index scoring, items were weighted on the standardized coefficients (average of two regressions) and revised according to the recommendations of three senior experts (in pediatrics, social care, and public health).

For the index validation phase, discriminant properties were assessed for two levels of deprivation: (i) moderate deprivation, defined as the need for any single type of specific health care required for deprived children; and (ii) severe deprivation, defined as the need for multidisciplinary care, consistent with admittance in PASS. For the index validation phase, the sensitivity and specificity were calculated with 95% confidence intervals.

The FrenChILD-Index reproducibility was assessed via a phone call retest, 5 to 6 months after the initial evaluation (period longer than schooling or health insurance application). This was applied on a random sample (0.3%) of all children initially included in the FrenChILD-Index study. It included children in a situation perceived as stable by parents and unadmitted to PASS over the period and was stratified with no more than 1/3 of children without any initial deprivation criteria.

Statistical analysis used the chi-squared, Fisher’s, and Student’s tests, Spearman and Pearson correlations, linear multiple regressions, bootstrap confidence interval calculation (using 1000 samples generated by unrestricted random sampling with stratification according to expert assessment: mean sample size = 624) with SPSS 20.0 (IBM, Armonk, NY, USA) and SAS 9.4 (SAS, Cary, NC, USA) software.

## 3. Results

This study shows the development of the FrenChILD-Index and its validation by resampling following the TRIPOD guidelines [45]. Flow chat of the study is displayed in Figure 1.

### 3.1. Item Generation Phase

In the item generation phase, 52 different items were extracted from seven published indices [28,29,31,46,47,48,49] and 13 senior expert interviews. Two indices were excluded because one concerned the underlying medical vulnerability of children [50], the other used informal interviews (no predefined questionnaire available), and the validity was not assessed [34]. The item generation and reduction are presented in Appendix A.

### 3.2. Item Reduction Phase

In the item reduction phase, 13 items were selected by the steering committee. Items covered the six SDH domains: were understandable despite low literacy; were helping to develop a care plan, obtain social support or improve family habits; and were acceptable in pediatric care (simple, closed, and easy to translate questions). The steering committee added an appendix for professionals, to provide elements for item justification and appropriate referral.

The reading group reordered items and judged 10 of the 13 items to be appropriate and three uncertain. There was no inappropriate item.

### 3.3. Index Derivation Phase

In the index derivation phase, the cross-sectional multicentric study analyzed 986 children. Their characteristics are presented in Table 1.

Four patients (three in center and one in center 2) out of the 990 initially included were excluded because of missing values (status about: French speaking for two, health insurance for one, and food security for one). The expert assessed that one was deprived in one domain (family context) and none had moderate or severe deprivation. The FrenChILD-Index is presented in Figure 2.

Item derivation including univariate and multivariate analysis is presented in Appendix A. The household vulnerability item was dropped because the reading group judged its appropriateness as uncertain, and it had no relevant correlation. Twelve items were included in multivariate analysis.

Factorial analysis found five dimensions, but the first dimension was overcoming all others. This justified that the derivation of the score continued considering, unidimensionally, the global burden of deprivation and used linear multiple regression analysis.

Senior experts corrected the FrenChILD item weighting to fulfill several logical conditions: (1) non-French speaking had an independent and long-lasting impact (recent migration weight was shared with non-French speaking) [51]; (2) single parenthood alone did not require assessment by a social worker; (3) homeless children or with incomplete health insurance and unknowing any social workers should be admitted to PASS [13,14]; and (4) homelessness was a wider determinant of health than the lack of health insurance as a higher weight.

The FrenChILD-Index had no ceiling effect. Scores ranged from 0 (428 (43.4%) children without any deprivation criteria) to 116 (one child with all the highest deprivation criteria), with a mean score of 13.1 (SD = 22.0), and a median score of 4 [interquartile range: 0; 15].

### 3.4. Index Validation Phase

In the index validation phase, expert assessment judged those 149 (15.1%) children needed at least one specific health care required for deprived children and 96 (9.74%) needed multidisciplinary care in a PASS.

The FrenChILD-Index correlated with the chosen proxies for global deprivation burden with the number of SDH domains affected by deprivation and the amount of specific health care required for deprived children 0.80 [95% CI: 0.77; 0.83] and 0.86 [95% CI: 0.83; 0.88] (*p* < 0.0005), respectively. The Kuder–Richardson Formula 20 coefficient was 0.76 [95% CI: 0.73; 0.78].

The cumulative frequency of the SDH domains affected by deprivation correlated with the FrenChILD-Index level (ρ = 0.67; *p* < 0.0005) (Figure 3).

A FrenChILD-Index ≥6 was screened for moderate deprivation with a sensitivity of 96.0% [95% CI: 92.6; 98.7] and specificity of 68.3% [95% CI: 65.2; 71.4]. A FrenChILD-Index ≥ 26 screened for severe deprivation with a sensitivity of 96.3% [95% CI: 92.7; 100] and a specificity of 91.1% [95% CI: 89.2; 92.9].

Among the children with severe deprivation, the 79 (45.9%) children not admitted to PASS had a lower FrenChILD-Index (*p* < 0.0005), more unhealthy housing (OR = 3.79 [1.55; 9.25]), and more single parenthood (OR = 2.15 [1.16; 4.01]). Those admitted to PASS were more frequently homeless, with unstable housing, unschooled, having recently migrated, non-French speaking, with no health insurance, in need of social assistance or not knowing how to get it, with missing medical records, or no health follow-up (*p* < 0.0005).

In the study center (2) with the lowest deprivation prevalence, the FrenChILD-Index retained suitable sensitivity and specificity for moderate deprivation (96.0% [95% CI: 89.8; 100]; 74.2% [95% CI: 74.6; 83.0]) and for severe deprivation (92.9% [95% CI: 82.1; 100]; 94.4% [95% CI: 92.0; 96.7]).

The FrenChILD-Index correlated with an expert assessment of children in every SDH domain except the family context. The risk of deprivation in the family context domain was higher (OR = 26.4 [95% CI: 10.8; 65.1]) for the FrenChILD-Index from 20 to 29. Experts associated family context deprivation with unhealthy housing (OR = 2.52 [95% CI: 1.39; 4.57]) and housing instability (OR = 3.19 [1.71; 5.98]), but not with homelessness (OR = 0.93 [0.12; 7.29]). They did not associate family context deprivation with interculturality (recent migration, *p* = 0.44; non-French speaking, *p* = 0.92).

The FrenChILD-Index retest for 31 children was highly correlated (r = 0.78; *p* < 0.01), and the mean difference was low (−0.71; SD = 7.51).

## 4. Discussion

The results showed that we have successfully constructed and internally validated an original and important pediatric individual-level index for deprivation in a European country. In the USA, IHELP had a similar sensitivity and specificity, but included items about adverse childhood experiences (limits discussed above) [32,52].

The small amount of missing data enabled a powerful multidimensional patient evaluation and showed a high admissibility and implementability. The FrenChILD-Index met all of Terwee’s criteria for index validation [53]:Content validity: The objectives, target population, and concept measured were clearly described and senior experts were involved in item selection. The FrenChILD-Index items significantly correlated with each corresponding SDH domain.Internal consistency: Items showed a good correlation with each other. The K20 value was over the acceptable threshold (>0.6) [41,42]. Linear regression enabled weighting items according to the proxies chosen to assess the progressivity of the global amount of deprivation. Other indices have already used this type of weighting [54,55,56].Criterion validity: Blinded cross-evaluation by an expert had already been used by Pascal and Colvin [38,52]. In clinical practice, experts are rarely available and one aim of the FrenChILD-Index was to replace them.Construct validity: The FrenChILD-Index showed how deprivation accumulates in SDH domains, causing early health inequalities [9,10]. Each item correlated with its SDH domain. Most were also correlated with the number of SDH domains affected by deprivation and the amount of specific health care required for deprived children (Appendix A).Individual interpretability: The FrenChILD-Index showed an excellent sensitivity and specificity for severe deprivation (≥26). Sensitivity was also excellent for moderate deprivation (≥6), but the specificity was slightly lower. Indeed, like with other screening instruments, intermediate situations were those in need of further expert evaluation. The FrenChILD-Index performances allows for efficient child referral to PASS or other types of support (Figure 2 and Appendix A). The appendix also provides additional information for appropriate referral, while its lack is known to be a barrier to screening [35,57].Collective interpretability: The FrenChILD-Index scaling has a sufficient number of degrees and covers every SDH domain (Table 2). This enables us to describe various deprivation levels and to show the proportionate impact on early health inequalities.Reliability: was good in the retest evaluation. This highlights the stability of the FrenChILD-Index over time.A floor but no ceiling effect: The FrenChILD-Index has a floor level because it focuses on deprivation. No ceiling effect means that there is no failure to discriminate between the highest levels. Thus, even though FrenChILD-Index has a maximum, it can describe a wide spectrum of deprivation levels.

Individual-level indices have to be raised beside area-based indices and they can better describe the potential SDH associated with less frequent diseases or conditions. Indeed, the higher number of predictors, more reliable themselves, recorded for each patient compensates for the small sample size in studies. These allow for reliable sample standardization in clinical research and in evaluating actions to reduce health inequalities.

The FrenChILD-Index, as an individual-level index, can describe individual specific needs and the articulation of complex care pathways, whereas area-based indices are inappropriate to describe them accurately. For severely deprived patients, combined interventions are needed to close the gap in SDH. Overtime, for moderately deprived patients, several interventions may also be needed. Studying the appropriateness of health care pathways is a new challenge to fit health services with the patient resources and needs [58,59].

Individual-level and area-based indices are also complementary, as shown by Stahlman et al. [3] and Nguyen et al. [21] in explaining several health inequalities. They help to understand which factors act as intermediates among several confounders in the different SDH domains.

Use of the FrenChILD-Index will improve the individual screening for SDH deprivation in clinical practice. Garg showed that systematic screening highly increases appropriate referrals [19]. However, physicians need resources to refer patients from mainstream health care to appropriate professionals, so this lack often hindered them from asking the questions [15]. Involving health care professionals in this screening enables them to ensure equity in their health care delivery.

The parents’ level of education was not retained by the experts. This single question has been used in several studies to assess SDH [6,21]. However, it does not cover all SDH domains (social surrounding, physical environment, access to health…). Furthermore, the experts judged that it was hard to ask about this during the health care of a child because this could be seen as discriminatory, and the professionals were reluctant to ask them about it because they could not justify it as necessary in order to consider an appropriate care or support action.

The FrenChILD-Index has raised the awareness of a potential expert judgment bias. Experts have reported less family context deprivation in cases of homelessness, but not in cases of interculturality. Indeed, even for the experts, homelessness raised questions about the lack of standards on how to be a good enough parent in such severely deprived living conditions.

This study had some limitations. The literature review started in 2013. However, it was regularly updated [31] and the study was continued in the absence of any satisfactory individual-level index. Although several experts were involved, interrater agreement among experts was not assessed due to organizational issues and the expected low patient admissibility. Three repeated unfunded interviews during a child’s stay were expected to lead to a response bias. Furthermore, Pascal also used single expert interview for reference in his study with a similar design in adults [38].

There were 172 children identified as severely deprived using the FrenChILD-Index, whereas experts retained referral to PASS for only 93. Indeed, PASS is a relevant resource for initial access to the health care system. However, severely deprived children need other types of social support (access to shelter, financial support, schooling…). Expert social assessment of patient needs is already a valuable type of care in complex living conditions [35]. Children with severe deprivation, according to the FrenChILD-Index but not admitted to PASS, matched some categories of Nicolas including some “acknowledged and well-managed economic deprivation” [34]. Furthermore, the FrenChILD-Index only gave three (0.3%) false negatives, highly appropriate in screening.

Response bias was individually possible (to get help or out of conformity). However, the experts were trained to cross-check and items uncorrelated with the expert’s opinion were dropped (i.e., household vulnerabilities). The FrenChILD-Index was tested in real-life settings of health care and has helped to start a dialogue with patients as well as sensitizing health care professionals to deprivation issues.

One hundred and fifty-one children were in situations of mild deprivation (FrenChILD-Index from 1 to 5). The use of PASS facilities did not permit us to validate this distinguishing of no versus mild deprivation.

Further studies including longitudinal cohorts are needed to fulfill validation of the FrenChILD-Index for mild deprivation in other health care facilities (e.g., mainstream maternal and child protection services) and provide more information about their needs and evolution. Determinism of health inequalities can now be measured between the different degrees of FrenChILD-Index and according to each item.

## 5. Conclusions

We developed and validated the FrenChILD-Index in response to the lack of any validated pediatric deprivation index in Europe. It is now used in France for appropriate individual referral and for epidemiological adjustment to deprivation biases and available for the stratification of children samples and the assessment of impact in public health interventions. It will improve the professional knowledge about the social determinants of health and the patients’ early health inequalities.

## Figures and Tables

**Figure 1 ijerph-19-16949-f001:**
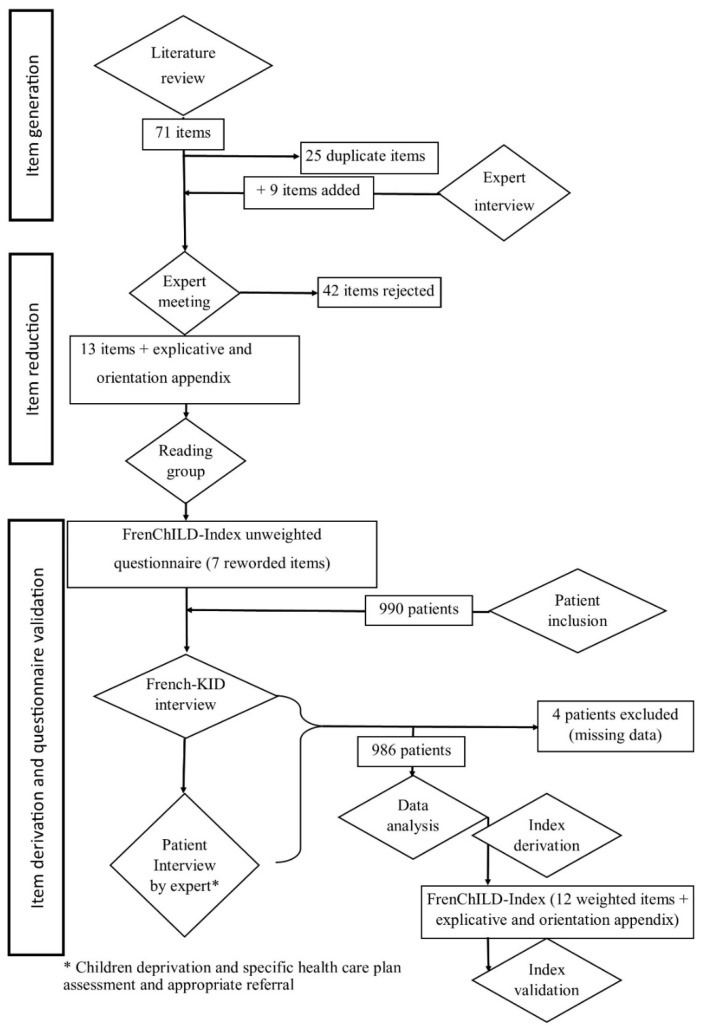
Study flowchart.

**Figure 2 ijerph-19-16949-f002:**
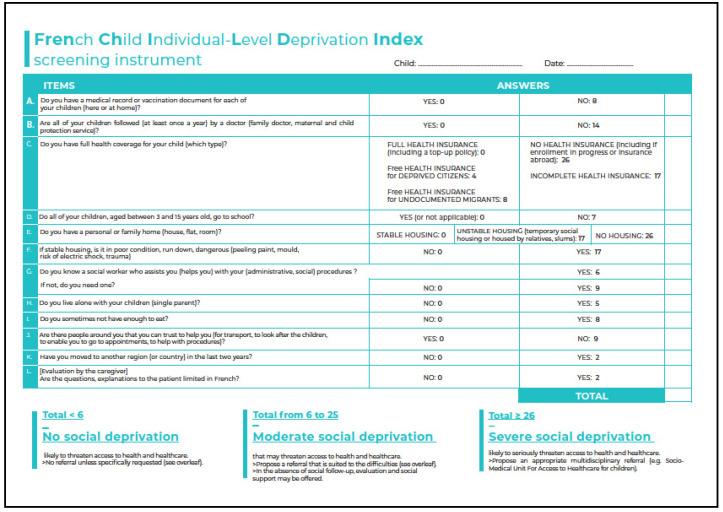
French Child Individual-Level Deprivation Index screening instrument (English translation).

**Figure 3 ijerph-19-16949-f003:**
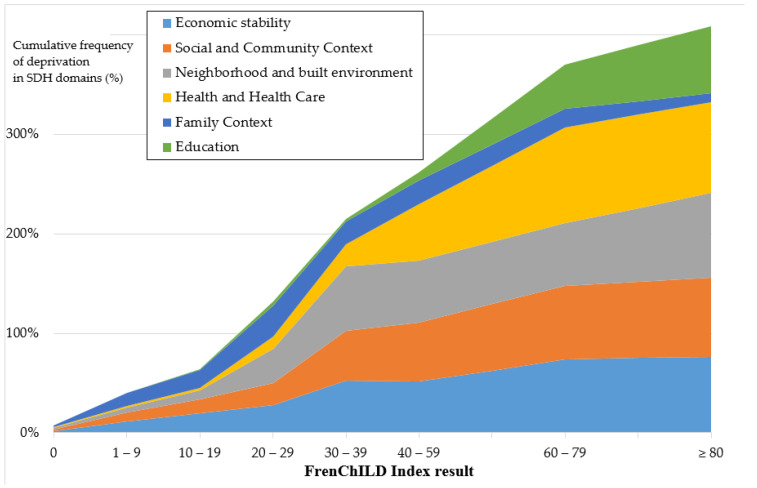
Piling up of the cumulative frequency of deprivation in the social determinants of health (SDH) domains by French Child Individual-Level Deprivation Index (FrenChILD-Index) level.

**Table 1 ijerph-19-16949-t001:** Characteristics of the children.

			Center 1		Center 2		*p*
			N	%	N	%	
			596		390		
	Sex (boys)		324	54.4%	210	53.8%	0.90
	Age, years *		8.60	(3.68)	8.65	(3.52)	0.84
**FrenChILD-Index items**							
(weighting to add up)		Answers					
**1. Do you have a medical record or vaccination document for each of your children (here or at home)?**							
(8)	Health booklet missing		38	6.4%	12	3.1%	<0.05
**2. Are all of your children followed (at least once a year) by a doctor (family doctor, maternal and child protection service)?**							
(14)	Children health follow-up missing		48	8.1%	18	4.6%	<0.05
**3. Do you have full health coverage for your child (which type)?**							
(0)	Full health insurance (including a top-up policy)		326	54.7%	306	78.5%	<0.0005
(4)	Complete health insurance for deprived citizens		182	30.5%	59	15.1%	
(17)	Incomplete health insurance		19	3.2%	9	2.3%	
(9)	Complete health insurance for undocumented migrants		14	2.3%	3	0.8%	
(26)	No health insurance (even if enrollment in progress or insurance abroad)		55	9.2%	13	3.3%	
**4. Do all of your children, school-aged ^a^, go to school?**							
(7)	Children unschooling		46	7.7%	9	2.3%	<0.0005
**5. Do you have a personal or family home (house, flat, room)?**							
(0)	Stable housing		536	89.9%	375	96.2%	<0.0005
(16)	Unstable housing (temporary social housing or housed by relatives, slums)		53	8.9%	10	2.6%	
(26)	Homelessness		7	1.2%	5	1.3%	
**6. If stable housing, is it in poor condition, run down, dangerous (peeling paint, mold, risk of electric shock, trauma)?**							
(17)	Unhealthy housing		54	9.1%	27	6.9%	0.24
**7. Do you know a social worker who assists you (helps you) with your (administrative, social) procedures? If not, do you need one?**							
(0)	Not needing a social assistance		429	72.0%	295	75.6%	<0.005
(6)	Needing social assistance		118	19.8%	83	21.3%	
(9)	Needing social assistance without knowing how to get this		49	8.2%	12	3.1%	
**8. Do you live alone with your children (single parent)?**							
(5)	Single parenthood		170	28.5%	85	21.8%	<0.05
**9. Do you sometimes not have enough to eat?**							
(8)	Food insecurity		57	9.6%	22	5.6%	<0.05
**10. Are there people around you that you can trust to help you (for transport, to look after the children, to enable you to go to appointments, to help with procedures)?**							
(9)	Surrounding dismiss		100	16.8%	25	6.4%	<0.0005
**Is there a particular personal situation at home (pregnancy, stress, chronic illness, disability, dependency, violence) that could affect your children’s life or health?**							
^b^	Household vulnerability		63	10.6%	33	8.5%	0.28
**11. Have you moved to another region (or country) in the last two years?**							
(2)	Recent migration		89	14.9%	34	8.7%	<0.005
**12. [Evaluation by the caregiver] Are the questions, explanations to the patient limited in French?**							
(2)	Non-French speaking		54	9.1%	9	2.3%	<0.0005
**Expert evaluation**							
**Domains of deprivation**							
	Health and Health Care		85	14.3%	22	5.6%	<0.0005
	Neighborhood and built environment		115	19.3%	32	8.2%	<0.0005
	Social and Community Context		35	5.9%	9	2.3%	<0.05
	Education		35	5.9%	9	2.3%	<0.0005
	Economic stability		134	22.5%	310	79.5%	<0.0005
	Family Context		91	15.3%	15	3.8%	<0.0005
Number of deprivation domains affected *			0.98	(1.40)	0.34	(1.03)	<0.0005
**Specific health care required for deprived children**							
	Free medical consultations		61	10.2%	13	3.3%	<0.0005
	Social evaluation to gain access to health insurance		72	12.1%	22	5.6%	<0.005
	Free medicine dispensing		60	10.1%	15	3.8%	<0.0005
	Home visit because of unhealthy housing		27	4.5%	29	7.4%	0.05
	Physical accompaniment in health care		17	2.9%	18	4.6%	0.14
	Multidisciplinary care coordination meetings		32	5.4%	25	6.4%	0.49
	Admission in PASS **		68	11.4%	28	7.2%	<0.05
Amount of specific health care required for deprived children *			0.45	(1.16)	0.31	(1.02)	0.05

* mean (standard deviation); ** PASS: specific medico-social units for access to health (Permanence d’Accès aux Soins de Santé); ^a^ Schooling was mandatory in France from the age of 3 to 15 from 1 September 2019. Previously, it was mandatory from the age of 6; ^b^ Item dropped after univariate analysis (see Appendix A).

**Table 2 ijerph-19-16949-t002:** Frequency of specific health care for deprived children per French Child Individual-Level Deprivation Index (FrenChILD-Index) level.

FrenChILD-Index	0		1 to 5		6 to 25		≥26	
Number of children (%)	428	(43.4)	151	(15.3)	235	(23.8)	172	(17.4)
**At least one specific health care required for deprived children**	2	(0.5)	4	(2.7)	17	(7.2)	126	(73.3)
**Admission in PASS ***	0	(0.0)	1	(0.7)	2	(0.9)	93	(54.1)
**Specific health care for deprived children**								
Free medical consultations	0	(0.0)	0	(0.0)	0	(0.0)	74	(43.0)
Social evaluation to gain access to health insurance	0	(0.0)	1	(0.7)	6	(2.6)	87	(50.6)
Free medicine dispensing	0	(0.0)	0	(0.0)	0	(0.0)	75	(43.6)
Home visit because of unhealthy housing	1	(0.2)	2	(1.3)	7	(3.0)	46	(26.7)
Physical accompaniment in health care	1	(0.2)	0	(0.0)	2	(0.9)	32	(18.6)
Multidisciplinary care coordination meetings	0	(0.0)	2	(1.3)	4	(1.7)	51	(29.7)

* PASS: Specific Medico-Social Units for Access to Health Care (Permanence d’Accès aux Soins de Santé).

## Data Availability

The data presented in this study are available in Appendix A.

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
