# Peer review of "Developing and Validating an Individual-Level Deprivation Index for Children’s Health in France"

_ijerph, 2022, doi:10.3390/ijerph192416949_

Round 1
Reviewer 1 Report
I found the article very interesting and greatly necessary. The authors have developed and validated the FrenChILD-Index. They have attempted to rigorously and systematically assess the individual level of deprivation for clinical practice. This is a pediatric index that is unique in Europe. The important thing is its great applicability and its use can be easily implemented in the European territory in a fast, solid and effective way. The index can allow clinical practice to rigorously incorporate the social aspects that determine health. That is why I believe that it is a very interesting instrument with a view to improving public and community health. The development and validation process seems adequate, rigorous and systematic. It complies with the most significant technical steps that exist for this type of methodological process. The tests carried out in the real world have seemed to me to be adequate and rigorous, this gives it a plus of effectiveness when it comes to generalizing its use at the individual pediatric level and at the level of public health.
The benefits of the instrument allow a more effective individual referral, a better epidemiological adjustment to deprivation biases, but it also serves to improve the knowledge of professionals about the social determinants of health and the inequalities that occur in early health. The limitations presented by the instrument are manageable and will improve with its daily use by professionals and experts in health centers. The literature review carried out, the methodological rigor in the index development and validation process and the systematic presentation of the results make this work an appropriate and significant contribution to its scientific field. My general assessment is very favorable to its publication.
Author Response
Dear reviewer,
We want to gratefully thank you for all your enthusiastic comments. They are very encouraging to the topic and work for our study. To our knowledge, you didn’t adress us any question. Revised manuscript will undergo final english editing before submission. But, as we understand, we had to respond to your comments first.
We hope you will be satisfied with the changes asked by other reviewers.
Best regards.
Reviewer 2 Report
An important contribution to science and to public policies. It should be accepted. But I add two suggestions, for some minor revision to make things clearer.
If I have well understood the paper (since I am not a statistician), the novelty of the index the authors have conceived will be to allow both broad epidemiological studies that may contribute to public policies and the identification of specific individuals, in hospitals or other health centers, that need more (or less) attention from the health professionals. I think the first point is very clear, but if I have well understood the paper, I think the second point should be emphasized.
Another comment is that there are many acronyms in English, but since they refer to French institutions it would be interesting to provide their names, and acronyms, also in French, or even mainly in French.
Author Response
Dear reviewer,
We want to gratefully thank you for all your enthusiastic comments. Please find below aswers to your comments :
- We translated the acronyms into French and explained their meaning: “Permanences d’Accès aux Soins de Santé” (PASS)
- We have emphasized in discussion the the contribution of FrenChILD-Index in identifiing individuals specific needs: « Individual interpretability: FrenChILD-Index showed an excellent sensitivity and specificity for severe deprivation (≥ 26). Sensitivity was also excellent for moderate deprivation (≥ 6) but specificity was slightly lower. Indeed like with other screening instruments, intermediate situations were those in need of further expert evaluation. FrenChILD-Index performances allows efficient child referral to PASS or other type of support (Figure 3, Figure S4). The appendix also provided additional information for appropriate referral, while its lack was known to be a barrier to screening [35,57].» «FrenChILD-Index, as an individual-level index, can describe individual specific needs and the articulation of complex care pathways, whereas area-based indexes are inappropriate to describe them accuratly. For severly deprived patients, combined interventions are needed to close the gap of SDH. But overtime, for moderatly deprived patients too, several interventions may also get needed. Studying healthcare pathways appropriateness is a new challenge to fit health services with patients' resources and needs [58, 59].
Individual-level and area-based indexes are also complementary, as shown by Stahlman et al. [3] and Nguyen et al. [21] in explaining several health inequalities. They help to understand which factors act as intermediate among several confounders in the different SDH domains.
Use of FrenChILD-Index will improve individual screening for SDH deprivation in clinical practice. Garg showed that systematic screening highly increases appropriate referrals [19]. Physicians need ressources for referring patients from mainstream healthcare to appropriate professionals, whereas this lack often hindered them asking the questions [15]. Involving healthcare professional in this screening enables them to ensure equity in their healthcare delivery. »
Revised manuscript will undergo final english editing before submission. But, as we understand, we had to respond to your comments first.
We hope you will be satisfied with the changes asked by other reviewers.
Best regards.
Reviewer 3 Report
This paper reports on the creation and validation of a deprivation index to assess child deprivation on multiple domains of the SDoH as applied to child patient rosters at two French hospitals. The paper is interesting and a welcome development in relation to other, area-based measures of deprivation that have been trialed across the EU and internationally now for decades. However, the paper suffers from a fairly major issue--that being the lack of clarity in methodological implementation--that significantly limits the communication and validity of the study results. I have included more detailed remarks below, but this would need to be significantly improved (especially the literature review) in order to be suitable for publication.
SPECIFIC COMMENTS
Title - it is grammatically incorrect. it's also not clear why the word 'daily' is included. Suggest "Developing and validating and individual-level deprivation index for children's health in France"
L53 - define multidisciplinary care
L61 - you define moderate deprivation, but not severe and or low deprivation? Please define
Somewhere in your introduction, you need to explain how the system of care works in France for the international readership of IJERPH - it’s not clear how medical consultations are paid for to the reader who is unfamiliar with the French system
METHODS - your lit review is already out of date by almost 6 years, calling into question many of the claims you make in your introduction around the lack of indices and validated tools. Your reader can’t take any of these comments at face value because you do not have an updated lit review. I see several recent studies (references 11/12) included beyond the year range, but other important contributions are omitted. See for example, Merville et al. 2022 IJERPH article on the F-EDI; or Guillaume et al.’s 2015 article in JECH
L76 - please define data saturation and how you know when you arrived at a saturation point following guidelines in the literature (see for example, Fusch et al 2015: https://scholarworks.waldenu.edu/facpubs/455/; Kerr et al. 2010 https://www.tandfonline.com/doi/abs/10.1586/erp.10.30; and Guest et al. 2020: https://journals.plos.org/plosone/article?id=10.1371/journal.pone.0232076)
Similarly, for line 80, please define consensus (e.g. simple majority? Some other threshold?)
Please also define the criteria utilized in considering the feasibility and inclusion/exclusion of your items to be reduced, as gauged by your 13 senior experts. The lack of process here calls into question the robustness of your method and therefore your results.
L83 - I’m not really sure what to make of your ‘independent reading group’… What is it that they did, exactly? What was their process for validation and confirmation of wording/measurement? Were they meant to be ‘reading’ the results in relation to the broader literature and available data? Please explain.
L88 - should be ‘convenience’ sample. Moreover, were you only selecting children experiencing deprivation? Or those without as a reference case? How many were recruited?
L89 - the wording here is confusing. You are talking about deprivation at an individual level throughout the article, but now apply the concept to hospitals? Please unpack this. I assume you are talking about broader neighbourhood deprivation?
L103 - is ‘global deprivation burden’ intended to be the resulting score of your FrenCHILD index? Please explain.
L107-109 - the way this is worded suggests practitioners are making a judgement call around ‘current social standards’. Can this be explained for your reader to improve the consistency of reporting and standardization of your data collection and assessment process?
L110 - This process is not clear. Why was a simple correlation analysis conducted and against what other data source? Why was a dimension reduction technique not used such as factor analysis to statistically ascertain the relevance of certain factors to SDoH?
L116 - did you not have a reference level for children not experiencing deprivation? Is that not one of the discriminant properties worth testing in relation to your two domains of deprivation?
L120 - this is not clear. Is this a random sample of those who already participated in the assessment? Another sample? What is the denominator for your reporting sample rate of 0.3%??? Moreover, is this meant to be your reference group? It would be helpful to more clearly unpack your methods and procedures for your reader. Based not his current draft, it is not entirely clear what was done…
L126 - please unpack your ‘analysis rules’
L130 - do you have neighbourhood level data to back up the claim of 25% deprivation? It currently looks like you are simply estimating this when it would be more reasonable to draw from the broader region(s) in question using empirical data
L134-Again, how are you correlating this data with a meta-level phenomenon like ‘education’ or ‘housing’? What are the data that are actually being matched here. Moreover, r of 0.3 is considered a weak association. Did you conduct a sensitivity analysis or examine what happens to your groupings when you increased the correlation to a reasonable value (e.g. higher than 0.5/0.6?)
L136 - Cronbach’s alpha is typically utilized when building a scale or an index. This has not been explicitly mentioned (beydon the idea that you are producing a meta-level deprivation index). Can you explain this? Are you building separate indices according to different domains of deprivation per different SDoH? Or is this a single index of deprivation? Again, can you also explain how your data were standardized so as to be relevant to applying Cronbach’s alpha, vs another technique such as factor analysis or PCA?
L139 - this makes it sound like you are correlating your proxy indicators of SDH domains with SDH domains affected by deprivation (tautological?) - it’s also not clear where your ‘domains’ are coming from and what the data sources are for each stage of your analysis, or what is meant by ‘domains of SDH’
Relatedly, your process and statistical methods could be better integrated to improve the flow of your methods section and improve clarity around the iterative process you have undertaken. Suggest revising.
L174 - I’m surprised you haven’t included any metrics around parental SES as a determinant of deprivation given its prominency in the literature. Can you explain?
L193 - please explain what you mean by ‘no ceiling effect’. If it’s an index, it would be bounded, no? How would a child experience an infinite score on the scale based on what was assessed and included? Please also comment in the discussion about the relatively low mean and median scores in relation to the types of care recommended for differently scoring children. For example, on L198 - how were these judged/assessed and in relation to what scores on the FrenChILD-Index?
L204 - noting Alpha is below typical threshold of 0.8
Figure 2 - please appropriate label your axis and ensure it is directed the appropriate way for your reader
Discussion - please return to the literature in a more compelling way to link and expand upon the value of an individual level vs. area level deprivation index in terms of what this adds that is not currently offered in existing tools. Please also more adequately address teh claim you make in your abstract that this is a tool that is useful for both clinical and public health settings. This is simply presented at face value and not unpacked in any meaningful way in terms of how both clinicians and public health practitioners could use this information to inform their practice. There are also other limitations of your study that are worth mentioning. For example, the cross-sectional collection of data says little about changing individual and neighbourhood levels of vulnerability and their influence on outcomes. Making recommendations for future enhancement and utilization of this tool would be appropriate here, especially given it is a new analysis technique and index.
Author Response
Dear reviewer,
We want to gratefully thank you for your comments. Our responses follow the order of your comments but the line numbers are not mentioned. This is because we have made major revisions and the revised manuscript will undergo a final English editing before submission. But, as we understand, we had to respond to your comments first. Please find enclosed the response to your comments.
We hope you will be satisfied with these changes and those asked by other reviewers.
Best regards.

Reviewer 4 Report
The work entitled “Daily tackling Health Inequalities: French Child Individual- 2 Level Deprivation Index development and validation” contains new scientific knowledge and covers a relevant topic. However, I have some comments to make that should be addressed before the manuscript could be considered for publication.
Introduction:
I would suggest authors to introduce more recent literature about the topic. For example, authors should specify what is the impact of deprivation on mental health (e.g. specific consequences).
Also, authors should comment about other instruments specifically devoted to measure deprivation (if any) or related phenomena.
Method:
Participants:
How did authors manage missing values?
Data Analysis:
In my opinion one of the main flaws of the article is that authors have not provided any information about evidences of validity of the internal structure and reliability of the scores. If the objective is to develop and validate an instrument a critical point is to assure that its internal structure is adequate. To this end the more accepted practice is to conduct firs an Exploratory Factor Analysis (EFA) and then a Confirmatory Factor Analysis (CFA). In addition, information about the measurement invariance of the instrument is relevant.
The fact that authors have not analyzed any of this aspect is critical for me in order to accept the document. As it appears at this moment.
Author Response
Dear reviewer,
We want to gratefully thank you for your comments. Please find answers below:
- We added in introduction the most recent literature we have found about deprivation:
« Social determinants of health (SDH) have a tremendous impact on children, generat-ing early health inequalities. Pillas et al. have widely reviewed the impact of SDH, on eve-ry organ during childhood, over 201 studies in European countries [1]. Pearce et al. have highlighted the long-lasting impact of SDH through the increase of overweight, poor mental wellbeing, longstanding illnesses, asthma and unintentional injuries [2]. SDH impact has recently increased over children’s and adolescents’ mental health, including sleep disorders [3, 4] and an impact over schooling [5]. Early health inequalities are also reinforced by barriers in access to healthcare determining vicious circles in therapeutic and preventive care [6,7].»
- We added in introduction the most recent literature we have found about other instruments specifically devoted to measure deprivation:
« Screening for deprivation is crucial to provide appropriate referral and support for chil-dren [19], and pediatricians often need assistance with this [20].
Area-based deprivation indexes have been implemented to compare deprivation within different geographic areas but were limited to study individual healthcare path-ways. Several studies have described health inequalities over frequent diseases in Europe for children in epidemiological studies [6,21,22]. However, strong variations in precision were found between area-based indexes [23] and between the components of an index [24]. Significant discrepancies were also found when compared with an individual-level index [25]. And Zelenina et al. are currently involved to review their properties [26].
However, most terms in these indexes (the rate of unemployment, of tenants, of single parenthood in an area…) are unavailable during care and poorly reliable to individual situations of care.
Several individual-level indexes were implemented in France. The “score de handicap social” used 111 items and couldn’t be realized in daily clinical practice [27]. Pascal et al. had only included questions about health insurance coverage and income [28]. EPICES had some inappropriate questions during child healthcare related to lifestyle confounding factors about leisure activities (going to show or on vacation) [29]. This variability was even more feared following the Covid. Fouchard compared three adult individual-level deprivation indexes and showed restrictions about their metrological performances [30]. Another index was designed for prenancy, but most items were very specific to this status [31].
In children, Sokol reviewed 11 indexes derived in the USA. Only 3 had validity as-sessed [8]. All these included items about adverse childhood experiences or parental de-pression. But screening for deprivation (socially considered to be a condition for assis-tance) and addressing adverse childhood experiences (socially banned and legally pro-hibited) should be considered separately because the latter are deemed to be judgmental and cause response biases [32]. WHO has also separated chapters of its strategy on social determinants of health reduction, and adverse childhood experiences prevention [33].
Nicolas et al. had used informal interviews to identify different situations of absolute precariousness requiring urgent social measures, recognized and well-managed precari-ousness, unrecognized precariousness and/or complicated by a significant deterioration in quality of life and psychological vulnerability. But validity of this interviews wasn’t evaluated [34].
Thus, to our knowledge, no convenient individual-level index to assess the global burden for child SDH deprivation has been validated in France or in Europe.»
- We clarified our management of missing data :
« Complete-case analysis was drawn to ensure there was no information bias of any esti-mator nor reference evaluation, thus patients missing any value were excluded.»
- We justified our methodology for internal validity:
«Internal consistency of the index was assessed using Kuder-Richardson Formula 20 coef-ficient (equivalent to Cronbach alpha for dichotomous items [41,42,43]). For derivation of the index, a linear multiple regression weighting was chosen to maximize the predictivity with respect to the progressiveness in global deprivation burden [44]. »
«Internal consistency: Items showed a good correlation with each other. Kuder-Richardson Formula 20 value was over acceptable threshold (> 0.6) [41,42]. Linear regression enabled weighting items according to the proxies choosen to assess the progressivity of global amount of deprivation. Other indexes had already used this type of wheighting [54-56]. »
Revised manuscript will undergo final english editing before submission. But, as we understand, we had to respond to your comments first.
We hope you will be satisfied with the changes asked by other reviewers.
Best regards.